# Mildly Impaired Foot Control in Long-Term Treated Patients with Wilson’s Disease

**DOI:** 10.3390/jfmk7010005

**Published:** 2021-12-31

**Authors:** Sara Samadzadeh, Harald Hefter, Osman Tezayak, Dietmar Rosenthal

**Affiliations:** 1Department of Neurology, University of Düsseldorf, Moorenstrasse 5, D-40225 Düsseldorf, Germany; sara.samadzadeh@yahoo.com (S.S.); tezayak@hispeed.ch (O.T.); dietmar.rosenthal@med.uni-duesseldorf.de (D.R.); 2Department of Psychiatry, Psychiatriezentrum Kreuzlingen, Nationalstrasse 19, CH-8280 Kreuzlingen, Switzerland

**Keywords:** Wilson’s disease, abnormal gait, reduction of gait speed, bradykinesia, ground reaction forces, optimization of therapy

## Abstract

Abnormal gait is a common initial symptom of Wilson’s disease, which responds well to therapy, but has not been analyzed in detail so far. In a pilot study, a mild gait disturbance could be detected in long-term treated Wilson patients. The question still is what the underlying functional deficit of this gait disturbance is and how this functional deficit correlates with further clinical and laboratory findings. In 30 long-term treated Wilson patients, the vertical component of foot ground reaction forces (GRF-curves) was analyzed during free walking without aid at the preferred gait speed over a distance of 40 m. An Infotronic^®^ gait analysis system, consisting of soft tissue shoes with solid, but flexible plates containing eight force transducers, was used to record the pressure of the feet on the floor. Parameters of the GRF-curves were correlated with clinical scores as well as laboratory findings. The results of Wilson patients were compared to those of an age- and sex-matched control group. In 24 out of 30 Wilson patients and all controls, two peaks could be distinguished: the first “heel-on” and the second “push-off” peak. The heights of these peaks above the midstance valley were significantly reduced in the patients (*p* < 0.05). The time differences between peaks 1 or 2 and midstance valley were significantly negatively correlated with the total impairment score (*p* < 0.05). Gait speed was significantly correlated with the height of the “push-off” peak above the midstance valley (*p* < 0.045). The GRF-curves of free walking, long-term treated patients with Wilson’s disease showed a reduced “push-off” peak as an underlying deficit to push the center of mass of the body to the contralateral side with the forefoot, explaining the reduction in gait speed during walking.

## 1. Introduction

Wilson’s disease (WD) is an autosomal recessively inherited disorder of copper metabolism [1,2]. The underlying genetical defect of the gene ATP7B is localized on chromosome 13 (13q14.3–q21.1) [3,4,5], affecting the regulation by an ATP7B encoded enzyme of the copper transport in the human body [3]. Reduced synthesis of ceruloplasmin and reduced biliary excretion of copper cause elevated serum levels of unbound copper and tissue accumulation of toxic copper in multiple organs during the course of untreated WD [2,6].

In the central nervous system (CNS), copper is predominantly stored in the grey matter of the basal ganglia and the cortex [7,8], resulting in typical neuropsychiatric manifestations [6,9]. However, MRI- and PET-scanning also indicate the additional involvement of the brainstem and cerebellar nuclei [7,8]. These structures contribute to the clinical spectrum of symptoms. Impaired eye movements [6,10] and impaired breathing [11] have been described, but not studied in detail.

Walking is another motor task that is influenced by basal ganglia, cerebellar, and brainstem nuclei. It is an essential motor skill and the first locomotion pattern to appear in children in an upright position [12] By about 3 years of age, lower limb joint motion during walking develops adult-like consistency [13]. As WD does usually not manifest before the age of five [6], it can be assumed that WD patients have learned to walk normally. In clinical practice, dystonic, ataxic, and parkinsonian gait patterns are distinguished in WD [14].

The prevalence of gait abnormalities among initial symptoms of WD ranges from not being present (0%) or being observed in <10% [15], <17% [16], <40% [17], <45% [14], up to 59% [18], and even 75% [8] of the cases. In late-diagnosed WD patients, gait disorders may become severe with frequent falls [18]. Using clinical scores, Burke et al. [14] were able to demonstrate that gait responded excellently to therapy in WD in comparison with other neurological symptoms and in contrast to dystonia in 85 patients who had been recruited in three different clinical trials to test the safety and efficacy of tetramolybdate [14]. This implies that, in long-term treated WD patients in contrast to de novo patients, gait disturbances will become rare and mild.

This was confirmed by clinical examination in long-term treated WD patients [19] and by the first quantitative gait analysis in WD [20]. This study measured the spatiotemporal relationship between both legs [20]. Evidence was presented that, in early diagnosed, long-term treated compliant WD patients, the gait disorder is mild, and the central pattern generator of walking (organizing the spatiotemporal time structure of bipedal interaction [21,22,23]) appears to be intact [20]. However, this study did not provide any explanation for the described slowness during free walking. On the basis of the frequently observed dystonia and bradykinesia during voluntary alternating finger movements [17,19], we hypothesize that WD patients also suffer from a subtle control deficit of foot and toe movements. The precise measurement of foot and toe movements is difficult using conventional gait measuring devices. Therefore, the vertical component of ground reaction forces (GRF-curve) was analyzed in the present study to detect subclinical hints for a mild forefoot control impairment during the push-off phase.

## 2. Materials and Methods

The study was performed according to the Declaration of Helsinki. Publication of anonymized clinical data after the informed consent of patients with WD was approved by the local ethics committee of the University. The presented data are part of the medical thesis of Osman Tezayak [24].

### 2.1. Patients and Controls

In the outpatient department of the Neurological University hospital in Düsseldorf (Germany), about 120 WD patients are seen on a regular basis. Every second Wednesday per month, about 10 WD patients present to monitor therapy of WD. During these routine visits, patients were informed of the purpose of this study. On average, 5 patients could be analyzed per Wednesday. Thus, the consecutive recruitment of the 30 outpatients in the present study could be completed within 6 months. Inclusion criteria were (i) long-term treatment of WD for more than 2 years and (ii) informed consent. Exclusion criterium was a history of walking deficits before the age of 5.

An age- and sex-matched control group was recruited from the hospital staff or relatives of the authors. All controls were healthy normal subjects with a normal neurological examination. Relatives of patients were not included to avoid recruitment of heterozygotic gen careers.

Demographical data (age, distribution of sex, and age at diagnosis) were documented and mass and body height were measured before the clinical neurological examination and gait analysis.

### 2.2. Clinical Examination and Scoring of Neurological Symptoms

Before gait analysis, patients and controls underwent a detailed clinical neurological examination. Seven motor symptoms (dystonia, dysarthria, bradykinesia, tremor, gait disturbance, oculomotor deficits, and ataxia of extremities), as well as three non-motor symptoms (reflex abnormalities, sensory symptoms, neuropsychological, and psychiatric symptoms), were scored on whether these symptoms were absent (0) or only mildly (1), moderately (2), or severely (3) present. The seven motor subscores were summed up to yield a motor score (MotS; 0–21), the three non-motor items were summed up to a non-motor score (N-MotS; 0–9), and the sum of MotS and N-MotS yielded the total score (TS; 0–30). These scores have already been used in previous studies [17,19,20]. A similar score is used by the Italian Monotematica AISF liver transplantation group [25]. Patients with TS <3 were classified as mildly, those with TS between 3 and 6 as moderately, and those with TS >6 as severely affected. In all control subjects, TS was zero.

### 2.3. Blood and Urine Analysis

Patients were treated with copper chelating agents. To monitor the efficacy and side effects of WD therapy, patients underwent a detailed analysis of blood and urine. Out of the long list of parameters used for routine WD therapy monitoring, the following parameters were selected for correlation with clinical scores and walking parameters: (1) parameters of copper metabolism (ceruloplasmin, serum copper, and the concentration of copper in the 24 h urine collected under medication), (2) liver enzymes (GOT, GPT, and GGT), (3) coagulation (thromboplastin time (PTT) and international normalized ratio (INR)), and (4) kidney function (serum level of creatinine (Crea)).

### 2.4. Gait Measurement

Before the recording of gait measurements was performed, the device was demonstrated to the patients. Measurements were only initiated after patients had given their consent a second time. All patients were able to walk without aid. None of the patients wore orthopedic shoes or an ankle-foot orthosis. Patients and controls had to walk a distance of 40 m at the preferred (natural) walking speed. Patients had to use their usual street shoes.

For gait measurement, the Ultraflex Infotronic^®^ system (NL-7650 AB Tubbergen, The Netherlands) was used (see Figure 1; for photos and further details, see http://www.infotronic.nl [26,27], accessed on 15 November 2021). It consists of light tissue shoes that can easily be strapped over the street shoes (Figure 1 left and right part). In the left part of Figure 1, a normal subject wearing the device is presented. In the solid but flexible plate of the Infotronic^®^ shoes, eight force transducers are integrated (Figure 1 middle parts), which record the vertical component of eight ground reaction forces. The ground reaction force (GRF) curve analyzed in the present study is the sum of the signals of these eight force transducers. For the determination of the onset and end of the stance phase, those timing points were chosen where the GRF-curve crossed the force level of 5% of peak amplitude.

For each single GRF-curve, the following amplitude parameters were extracted (see Figure 2): the amplitude of peak 1 (P1A), the amplitude of the midstance valley (V1A), and the amplitude of peak 2 (P2A). Furthermore, the difference from P1A to V1A (PD1), the difference from P2A to V1A (PD2), as well as the corresponding mean value (PD; PD1 + PD2)/2 (=P1A + P2A)/2 − V1A were calculated. Further temporal parameters were as follows: time to peak 1 (TP1), time to valley 1 (TV1), and time to peak 2 (TP2).

For each patient and control subject, the mean values across all GRF-curves of these nine parameters were calculated, as well as the nine corresponding intraindividual standard deviations (which were labeled P1ASD, V1ASD, P2ASD, PD1SD, PD2SD, PDSD, TP1SD, TV1SD and TP2SD).

Then, the mean values across all subjects in the patient and the control group were calculated for all nine parameters (P1A, V1A, P2A, PD1, PD2, PD, TP1, TV1 and TP2) as well as for the nine standard deviations (P1ASD, V1ASD, P2ASD, PD1SD, PD2SD, PDSD, TP1SD, TV1SD and TP2SD).

For further analysis, the following parameters were also selected: DUR = duration to walk the distance of 40 m in seconds; GSP = gait speed in m/s = 40 m/Dur; CAD = steps/s and number of steps (NUM) and step length needed.

### 2.5. Statistics

As some of our parameters were derived from discrete scales (e.g., MotS, N-MotS, and TS), we thus decided to perform non-parametric tests only. Group comparisons were performed using Wilcoxon’s test. For the correlation analysis, Spearman’s rho (rank correlation) was used. All tests were part of the commercially available SSPS statistics package (version 25; IBM Analytics, Armonk, NY, USA).

## 3. Results

### 3.1. Comparison of Demographical Data of WD Patients and Control Subjects

Patients were perfectly sex-matched by the controls: 19 patients were males and 11 were females. There was no significant difference in age (WD: mean age: 34.2 years, SD: 11.0; controls: mean age: 32.8 years, SD: 8.3; *p* = 0.571), body mass (WD: mean body mass: 72.7 kg, SD: 15.0; controls: mean body mass: 73.6 kg, SD: 14.8; *p* = 0.73), and body height (WD: mean body height: 177.8 cm, SD: 10.3; controls: mean body height: 176.9 cm, SD: 9.5; *p* = 0.83). Mean age at diagnosis ranged from 11 to 36 years (mean: 22.0 SD: 6.8 years) and duration of treatment from 31 to 376 months (mean: 144 SD: 106 months).

### 3.2. Clinical Gait Abnormalities in the Patient Cohort

Only 6 out of the 30 WD patients (20%) presented with a gait disorder during clinical neurological examination: two patients suffered from a typical parkinsonian gait (TS = 12 and TS = 7), three patients from a mild dystonic gait (TS = 7, TS = 7 and TS = 4), and one patient from an ataxic gait with the highest score (TS = 16). Four of these six patients did not have two distinguishable peaks in the GRF-curves (see below).

### 3.3. Recording of GRF-Curves in Differently Affected Patients

In Figure 3, the sequences of ground contacts of the left (L) and right foot (R) of a normal subject (left side), a moderately affected patient (TS = 5; middle part), and a severely affected patient (TS = 12; right part) are presented.

In Figure 4, the GRF-curves of the left (upper part) and the right foot (lower part) of the same subject and patients as in Figure 3 are superimposed with the time normalized stance phase. The more affected the patient, the slower he walks, the higher the variability of the individual steps, and the less clear the “heel-on-midstance-toe off” modulation of the CRF-curves. In the severely affected patient (Figure 3 and Figure 4 right side), the typical shape of the GRF-curve with two distinguishable peaks is missing.

### 3.4. Percentage of WD Patients with an Abnormal Shape of GRF-Curves 

In normal subjects, two peaks (P1 and P2) of the GRF-curves can clearly be distinguished (see Figure 3 and Figure 4 left side). Only six WD patients (=20%) did not have two peaks in their GRF-curves. The data of these six patients are presented in Table 1.

These six patients had significantly higher TSs and N-MotSs compared with the 24 patients with normally shaped GRF-curves (mean N-Mots in the 24 WD patients: 0.7, SD: 1.2; *p* < 0.03; mean TS in the 24 WD patients: 4.0, SD: 3.0; *p* < 0.03). Four of these six patients were classified with abnormal gait during clinical examination.

### 3.5. Walking Speed and Cadence in WD-Patients and Control Subjects

WD patients needed significantly (*p* < 0.001) more time than the controls to walk a distance of 40 m (DUR: WD patients: mean: 30.0 s, SD: 4.2, controls: mean: 26.9 s, SD: 1.7). The cadence and the gait speed were significantly (*p* < 0.001) lower (CAD: WD patients: mean: 1.81/s, SD: 0.14; controls: mean: 1.91/s, SD: 0.09; GSP: WD patients: mean: 1.35 m/s, SD: 0.17; controls: mean: 1.49 m/s, SD: 0.09). However, the number of steps (NUM) and step length needed to walk 40 m did not differ significantly between patients and controls. Gait speed showed a significant negative correlation (r = −0.461, *p* < 0.012) with TS, and a significant negative correlation (r = −0.547; *p* < 0.001) between all three items of N-MotS and N-MotS itself.

### 3.6. Analysis of GRF-Curves in 24 WD Patients with a Normal Shape

The mean peak amplitudes P1A of the right and left leg did not differ significantly, neither in the WD patients nor in the controls. For the amplitude of peak 1 (P1A, see Table 2), the amplitude of peak 2 (P2A), and the amplitude of midstance valley (V1A), there was a tendency to lower values in the 24 WD patients, with two distinguishable peaks in their GRF-curves. The differences PD1 and PD2 (see Table 2) were significantly lower in the WD-patients. The time to peak 1 (T1) did not differ between patients and controls. However, the variability of T1 (T1SD, *p* < 0.02) and variability of time to peak 2 (T3SD, *p* < 0.001) were significantly higher in the patients. The time differences (TV1–TP1 and TP2–TV1) tended to be shorter in the patients (see Table 2). In summary, the GRF-curves of the patients revealed a less clear modulation: peak 1 and 2 were lying closer together and the valley V1 was less deep.

Gait speed did not correlate with the difference PD1 (r = 0.339; n.s.), but did correlate with PD2 (see Figure 5; r = 0.412; *p* < 0.045) and the mean value (PD1 + PD2)/2 (r = 0.438; *p* < 0.032).

There was a significant negative correlation of PD1 with the item “sensory symptoms” (r = −0.519, *p* < 0.009) and of the intraindividual variability of peak 1 (P1ASD) with the item “cerebellar symptoms” (r = −0.402; *p* < 0.02).

### 3.7. Correlation of the Analysis of GRF-Curves with Clinical and Laboratory Findings

There was a significant negative correlation (r = −0.442; *p* < 0.03) between the time differences TV1–TP1 and TP2–TV1 (see Table 2) and the total score (TS). No correlations between clinical scores and amplitude measurements were found. There was a clear negative correlation between the serum level of thrombocytes and the intraindividual variability of time to peak 1 (T1SD; r = −0.543, *p* < 0.002), which is hard to interpret.

## 4. Discussion

WD patients walked slower than control subjects and had less clearly modulated GRF-curves. Especially, the height of the “push-off” peak PD2 over the midstance level was reduced and correlated with gait speed.

### 4.1. Severity of Symptoms in the WD Patients

All 30 WD patients in the present study were recruited from outpatients, were treated for 12 years on average, and responded well to therapy. Compared with WD patients in a study on initial gait disturbances [18], patients in the present study were diagnosed about 14 years earlier on average, and thus were less affected and presented with a low prevalence of clinical manifest gait abnormalities (20%) [14,18].

### 4.2. Abnormal Shape of the GRF-Curve in WD Patients

So far, no other study on GRF-curves in WD patients is available. Six of the 30 WD-patients did not have two distinguishable peaks in their GRF-curves (compare Figure 3 and Figure 4, right side). Compared with the remaining 24 patients, these 6 patients had significantly (*p* < 0.03) higher non-motor (N-MotS) and significantly (*p* < 0.03) higher total scores. Therefore, the loss of “heel-on-midstance-push-off” modulation of the GRF-curve is associated with increased severity of WD. Four of these six patients were also classified with abnormal gait during clinical examination.

### 4.3. Analysis of GRF-Curves with Two Distinguishable Peaks

The majority (80%) of the WD patients presented with a normal shape of the GRF-curves with two distinguishable peaks. However, in these 24 patients, the height of the “heel-on” peak (PD1) and the “push-off” peak (PD2) above the midstance level was also significantly lower than in the controls (see Table 2). The time difference between the time to “heel-on” peak TP1 and the midstance time (TV1), as well as the time difference between TV1 and time to “push-off” peak (TP2), were significantly (*p* < 0.03) negatively correlated with the total score TS. This indicates that, the less clearly the GRF-curve of a patient was modulated, the more he was clinically affected.

However, GRF-curves of WD patients were not only less clearly modulated, but also showed a much higher variability. The time to the passive “heel-on” peak TP1 showed a much higher variability (TP1SD) in the patients (*p* < 0.02) than in the controls. The intraindividual variability of the time to the active “push-off” peak TP2SD was even more significantly (*p* < 0.001) increased compared with the normal subjects. There was a significant correlation between variability of the “heel-on” peak amplitude P1ASD and the presence of cerebellar symptoms during clinical examination, indicating a cerebellar impairment of the control of the “push-off” peak.

### 4.4. The Interaction between Abnormalities of the GRF-Curve and Gait Speed

The tendency to lower values of the amplitude of the “heel-on” peak, the midstance level, and the “push-off” peak can be explained by the significantly reduced gait speed. However, the reduction in the modulation of the GRF-curve cannot be explained simply by the reduced gait speed. Interestingly, no correlation between the height of the passive “heel-on” peak above the midstance level with gait speed was found, which we would have expected if the reduction in gait speed had been the major cause for the reduced modulation of the GRF-curve. Instead, a significant (*p* < 0.045) negative correlation between gait speed and the height of the active “push-off” peak was found. Prolonged time to peak of the “push-off” peak and a reduced PD2 is clearly demonstrated in Figure 3 (middle part) for the moderately affected patient. We think that this is the underlying causal deficit for the slowness during free walking in WD patients. Because of the reduced ability to push the center of mass to the other side and forward, gait speed is reduced.

This is in full agreement with previous studies demonstrating a reduced rate of alternating finger movements and a prolonged time to peak of a force pulse generated with the index finger extensor in WD [17]. Further, it is consistent with the frequent finding of a mild to moderate bradykinesia of hand and arm movements during clinical investigation [19]. Whether this reduction of forefoot pushing also reflects the presence of a mild foot or toe dystonia is difficult to measure and nearly impossible to see during clinical examination.

In the WD patients, cadence (CAD) was also mildly but significantly reduced. Reduction in cadence is well-known for patients with hepatic encephalopathy [28,29]. However, none of the WD patients in the present cohort suffered from clinically relevant liver disease and none from hepatic encephalopathy. Furthermore, even in the severely affected patients, the gait was symmetric and only mildly disturbed compared with patients with an asymmetric gait after stroke [30,31]. Untreated WD patients may suffer from osteomalacia [6]. However, in long-term treated WD patients, bone deformities seem to be rare. Nevertheless, we recommend that especially older WD patients should be seen by an orthopedist.

## 5. Conclusions

The measurement of the vertical component of ground reaction forces during free walking demonstrates a subclinical impairment of foot control during free walking in long-term treated WD patients. The active “push-off” peak at the end of the stance phase and, therewith, gait speed is reduced. However, this reduction in gait speed is only a mild deficit, especially as the temporal structure of walking, which is generated by the central pattern generators of walking [20,23], appears to be unimpaired. Most of the relevant gait abnormalities in WD can already be detected during careful clinical investigation [19].

## 6. Limitations of the Study

Data recording in the present study was performed by means of the Infotronic Ultraflex^®^ system. This system has several obvious shortcomings. Normalization of the GRF-curves to body weight would probably have more clearly shown the correlation between reduced peak modulation and reduced gait speed. This rescaling of the force data delivered by the Infotronic^®^ system would have implied a completely new data analysis. Furthermore, normalization of an entire gait cycle would have allowed to analyse the interaction between both legs more clearly, but the Infotronic^®^ system only normalizes the stance phase for each foot separately.

A combination of angle and ground reaction force measurement of foot and toe movements would probably have more clearly shown the foot control deficit in long-term treated WD patients. However, even with the use of modern foot models, these measurements are difficult to perform, especially with systems using active markers. Furthermore, our patients were outpatients with limited time for recording, because gait analysis was combined with the analysis of hopping and running on the same day. Therefore, the analysis of GRF-curves recorded by means of the Infotronic Ultraflex^®^ system was as a compromise, as it could be performed easily and quickly.

## Figures and Tables

**Figure 1 jfmk-07-00005-f001:**
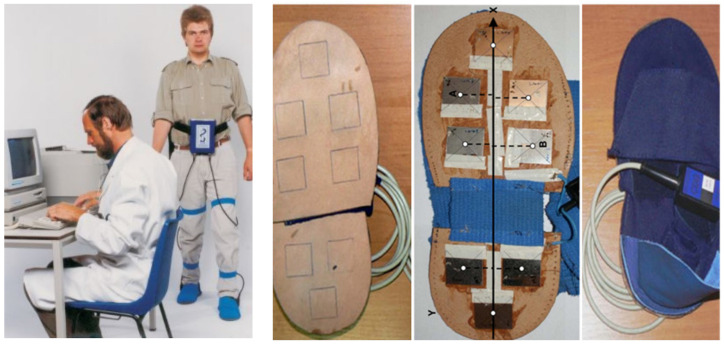
(**left part**) A normal subject wearing the shoes and the microprocessor recording the signals of the eight force transducer signals. (two middle parts) Localization of the eight transducers in the plate of the shoes. (**right part**) A pair of shoes of the Ultraflex Infotronic^®^ system.

**Figure 2 jfmk-07-00005-f002:**
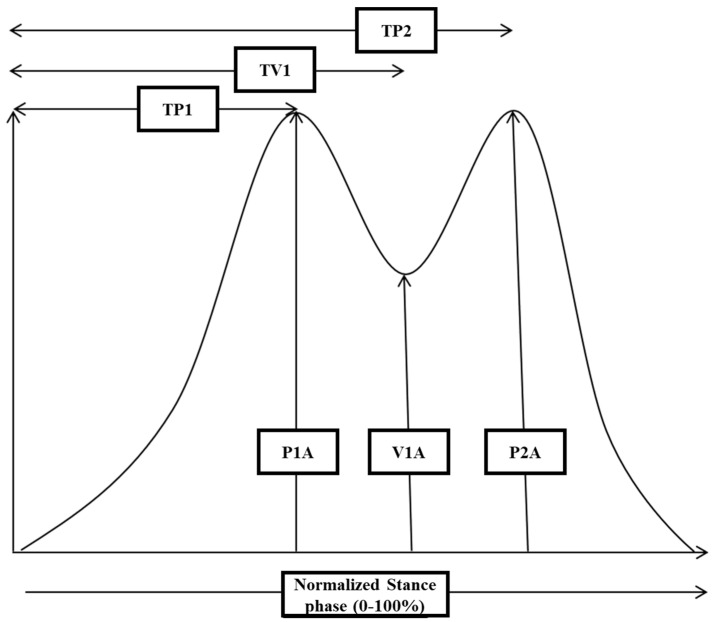
A schematic drawing of a normal GRF-curve is presented to demonstrate which parameters were extracted for further analysis: the amplitude of peak 1 (P1A), the amplitude of valley 1 (V1A), and the amplitude of peak 2 (P2A). To determine the onset and end of the stance phase, those timing points were chosen where the GRF-curve crossed the force level of 5% of peak amplitude. Thereafter, time to P1 (TP1), time to V1 (TV1), and time to P2 (TP2) were determined (for further details, see Section 2).

**Figure 3 jfmk-07-00005-f003:**
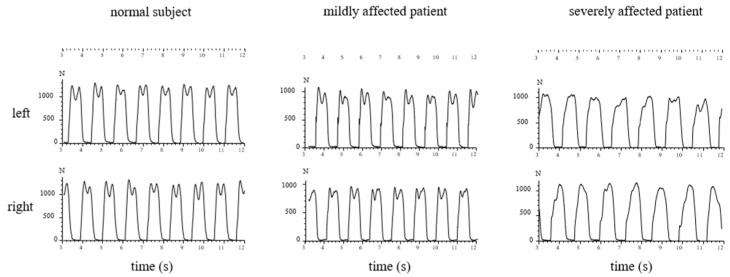
The GRF-curves of a few steps of a normal subject (**left side**), of a moderately affected WD-patient (**middle**), and of a severely affected WD-patient (**right side**) are presented. In the GRF-curves of the normal subject, two peaks can clearly be distinguished (left side). The GRF-curves of the moderately affected patient show a higher first “heel-on”peak, but a lower “push-off”peak (**middle part**). In the severely affected patient, the clear modulation of the GRF-curve with two peaks and the midstance valley is missing (**right side**).

**Figure 4 jfmk-07-00005-f004:**
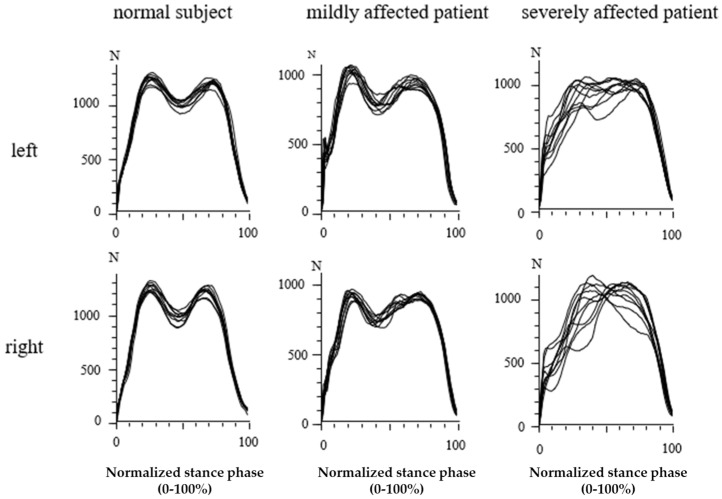
Superposition of the GRF-curves of the same subject and patients as in Figure 3 after time normalization of the stance phase. The GRF-curves of the normal subject are highly reproducible (**left side**) with two peaks. In the moderately affected patient (**middle part**), the variability of the GRF-curves is much higher. In the severely affected patient (**right side**), no peaks can be distinguished anymore.

**Figure 5 jfmk-07-00005-f005:**
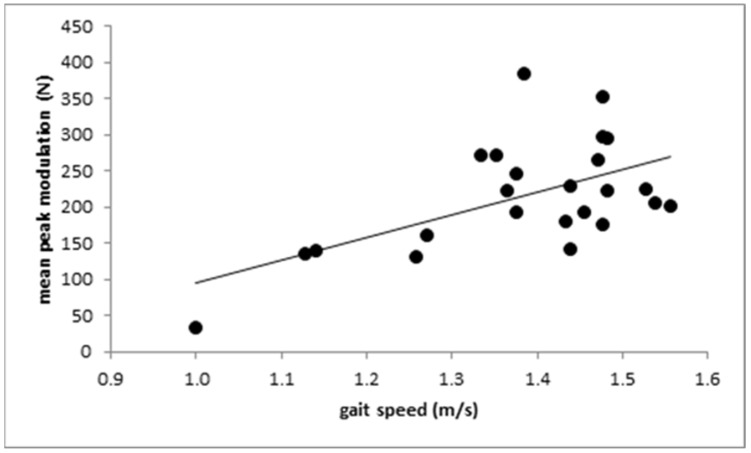
The positive correlation between the height of the “push-off” peak PD2 above the midstance amplitude V1A and gait speed.

**Table 1 jfmk-07-00005-t001:** Individual data of the six patients without two clearly distinguishable peaks in their GRF-curves.

Patient	Sex	Age	Bodymass (kg)	Body Height (cm)	MotS	N-MotS	TS
patient 1	f	56	60	164	10	2	12
patient 2	f	48	59	175	4	3	7
patient 3	m	15	60	188	4	2	6
patient 4	m	29	81	180	3	1	4
patient 5	f	40	58	172	2	1	3
patient 6	f	37	57	158	12	4	16
mean		37.5	62.5	172.8	5.8	2.2	8.0
SD		13.1	8.3	9.9	3.8	1.1	4.6

MotS = motor score; N-MotS = non-motor score; TS = total score; f = female; m = male; mean = mean value; SD = standard deviation.

**Table 2 jfmk-07-00005-t002:** Comparison of amplitudes and temporal parameters of the GRF-curves in 24 WD patients and 30 controls.

	WD Patients with Two Peaks (n = 24) Mean (SD)	Control Subjects (n = 30) Mean (SD)	Significance *p*-Value
P1A (N)	1060 (258)	1181 (297)	*p* < 0.12 (n.s.)
V1A (N)	775 (219)	801 (221)	*p* < 0.12 (n.s.)
P2A (N)	923 (241)	1000 (225)	*p* < 0.23 (n.s.)
PD1 (N)	285 (117)	380 (158)	*p* < 0.02
PD2 (N)	147 (75)	200 (102)	*p* < 0.04
TP1 (ms)	153 (16)	146 (16)	*p* < 0.13 (n.s.)
TP1SD (ms)	17 (15)	10 (4)	*p* < 0.02
TP2 (ms)	456 (53)	456 (29)	*p* = 1.0 (n.s.)
TP2SD (ms)	29 (13)	19 (11)	*p* < 0.001
TV1–TP1 (ms)	159 (27)	168 (23)	*p* < 0.18 (n.s.)
TP2–TV1 (ms)	144 (55)	142 (21)	*p* = 0.93 (n.s.)

Mean = mean value; SD = standard deviation; n.s. = not significant; P1A = amplitude of peak 1; V1A = amplitude of midstance valley 1; P2A = amplitude of peak 2; PD1 = difference between P1A and V1A; PD2 = difference between P2A and V1A; TP1 = time to peak 1; T1SD = intraindividual standard deviation of TP1; TP2 = time to peak 2; TP2SD = intraindividual standard deviation of TP2; TV1 = time to valley 1; TV1–TP1 = time difference between TV1 and TP1; TP2–TV1 = time difference between TP2 and TV1.

## Data Availability

Data available on request due to restrictions, e.g., privacy or ethical. The data presented in this study are available on request from the corresponding author.

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
