# Peer review of "Mildly Impaired Foot Control in Long-Term Treated Patients with Wilson’s Disease"

_jfmk, 2021, doi:10.3390/jfmk7010005_

Round 1
Reviewer 1 Report
The present study concern the gait abnormalities in 30 WD patients (measurement of the vertical component of ground reaction forces during free walking). The study is novel, based on a fairly large group of patients.
I have only 2 comments on methodological part
1. please characterize the control group - what kind of people were they, healthy, sick etc.
2. please describe why only 30 patients out of 120 were recruited, i.e. exclusion criteria.
3. I would also like to remind that for clinical exam UWDRS scale can be used, which was described in literature. This obviously does not affect the quality of the paper.
Author Response
The control group is now described in more detail.
Recruitment is now explained in more detail.
One of the authors (HH) has contributed to the development of the UWDRS. But in clinical practice we prefer our scale because it can be completed within 1 minute. This is not the case for the UWDRS, which collects a lot of redundant information.

Reviewer 2 Report
Mildly impaired foot control in long term treated patients with Wilson’s disease
The subject is pertinent, but the manuscript needs several major improvements.
Several parts are very difficult to follow due to incorrect English usage. For example:
L10: “observed initial symptom of Wilson´s disease, which, well to therapy, but has not been analyzed” -> “which, well to therapy”?; maybe “which respond well to therapy”?
L53-5: “ranges from not being present or not being observed over 10% [15] ], 17% [16]…”?; maybe “ranges from not being present (0%) or being observed in <10% [15] ], <17% [16] (… ) <75% [8] of the cases.”?
L69: “this study” – in this paragraph you have not presented any study
L214-5: “about 14 years earlier in the mean”, should be “about 14 years earlier on average”
Not “neurological investigation” but “neurological exam”
L260: is “mild foot” a “club foot”?
Please ask someone for help on the English writing.
You often use the term “laboratory findings” that can be any laboratory findings, but you are referring to “blood analysis”. Please replace “laboratory findings” by one form that is objective.
You present one Table separated in 3 parts, but by doing this you are presenting the results during the Materials and Methods section. Furthermore, the Table becomes too complex to understand.
Please separate into 3 different Tables and put each one in the adequate place.
The last column of your Tables is not the “level of significance” but the “p-value”.
Please put in “2.5 Statistics” the sentence: “The level of significance was set as alpha=0.05”, where “alpha” is to be replaced by the Greek letter alpha (I believe you use the common 0.05 value)
Throughout the manuscript, replace “P<…”, “P=…” with capital “P” by the same expression with lowercase “p”. Do not forget to include the “0” before the decimal point: “p<0.12” and not “p<.12”
Since your “WD-patients without two peaks” group has only six subjects, please show their individual values of mass, height, MotS, N-MotS, TS, in an anonymous way.
Also, in the Tables and the text, use mean+-SD, OR mean (SD).
If you use mean+-SD, then +- is to be replaced by one single character, the Plus-Minus Sign, with code 00B1 (hex) of Unicode.
In Table 1, you show the “body mass (kg)”, not the “body weight (kg)”
The entry for the WD patients is clearly wrong, you cannot have SD=1.5kg, for a range 48-126kg.
There is an extra B for the MotS entry (3.B3)
For T1 you show 153+-16 AND T1SD 17+-15. If you show mean+-SD, why do you have two entries for the SD? Furthermore, they are not consistent, the first is 16, while the second is 17. The same is true for T3.
L130-1: consider replacing “T1” by “TP1”, “T2” by “TV1” and “T3” by “TP2”.
L133 and elsewhere, there is no “secs”, you use either “s” or “second”.
It is not clear how you go from 8 mini-platforms curves to the complete curve, as show in figure 3.
Please show a similar figure, but with the curves for each of the 8 platforms plus the sum of them.
Show them large enough for the ready to see the summing process.
L121: what is “a center curve can be calculated yielding the final GFz-curve”? Please show a plot of it and how you compute it. The right part of figure 1 is not very clear.
In figure 1 (or a new figure) please show a subject wearing the device.
Please check the company address and internet site, as the link shown appears as “for sale”.
Has is usual in Biomechanics, the gait cycle comprises the stance (support) and the swing (aerial) phases. Therefore, please update your manuscript to these common definitions. For instance, in figures 2 and 4 the horizontal axis is not labelled correctly.
Also usual is to abbreviate “Ground Reaction Force” by “GRF”, not “GF” as you do. Please change.
And it is common to normalize the GRF to the Body Weight (BW). In this case, the values do not depend on the weight of the subjects, as will be the case for values expressed in Newton: the 126kg subject will have values more than two times the values for the 48kg subject, just by the weight difference.
Please show the GRF values in BW.
The relation between the values of the peaks and valley to the gait speed will become apparent.
Although figure 4 is pertinent, it will be far more instructive to have a figure with the left and right feet simultaneously, i.e., put the curve of figure 3 in just one row. In this way the double support phases will be clearly seen, and their relation to the walking speed.
The results shown in figure 5 are not the same as some mentioned on the paragraph L175-182.
L267: “the gait was symmetric” how do you quantify this symmetry?
Somewhere in the text you must say why you do not use statistical parametric tests.
Author Response
|
Reviewer 2 |
|
|
Mildly impaired foot control in long term treated patients with Wilson’s disease
The subject is pertinent, but the manuscript needs several major improvements.
Several parts are very difficult to follow due to incorrect English usage. For example:
L10: “observed initial symptom of Wilson´s disease, which, well to therapy, but has not been analyzed” -> “which, well to therapy”?; maybe “which respond well to therapy”?
L53-5: “ranges from not being present or not being observed over 10% [15] ], 17% [16]…”?; maybe “ranges from not being present (0%) or being observed in <10% [15] ], <17% [16] (… ) <75% [8] of the cases.”?
L69: “this study” – in this paragraph you have not presented any study
L214-5: “about 14 years earlier in the mean”, should be “about 14 years earlier on average”
Not “neurological investigation” but “neurological exam”
L260: is “mild foot” a “club foot”?
Please ask someone for help on the English writing.
You often use the term “laboratory findings” that can be any laboratory findings, but you are referring to “blood analysis”. Please replace “laboratory findings” by one form that is objective.
You present one Table separated in 3 parts, but by doing this you are presenting the results during the Materials and Methods section. Furthermore, the Table becomes too complex to understand.
Please separate into 3 different Tables and put each one in the adequate place.
The last column of your Tables is not the “level of significance” but the “p-value”.
Please put in “2.5 Statistics” the sentence: “The level of significance was set as alpha=0.05”, where “alpha” is to be replaced by the Greek letter alpha (I believe you use the common 0.05 value)
Throughout the manuscript, replace “P<…”, “P=…” with capital “P” by the same expression with lowercase “p”. Do not forget to include the “0” before the decimal point: “p<0.12” and not “p<.12”
Since your “WD-patients without two peaks” group has only six subjects, please show their individual values of mass, height, MotS, N-MotS, TS, in an anonymous way.
Also, in the Tables and the text, use mean+-SD, OR mean (SD).
If you use mean+-SD, then +- is to be replaced by one single character, the Plus-Minus Sign, with code 00B1 (hex) of Unicode.
In Table 1, you show the “body mass (kg)”, not the “body weight (kg)”
The entry for the WD patients is clearly wrong, you cannot have SD=1.5kg, for a range 48-126kg.
There is an extra B for the MotS entry (3.B3)
For T1 you show 153+-16 AND T1SD 17+-15. If you show mean+-SD, why do you have two entries for the SD? Furthermore, they are not consistent, the first is 16, while the second is 17. The same is true for T3.
L130-1: consider replacing “T1” by “TP1”, “T2” by “TV1” and “T3” by “TP2”.
L133 and elsewhere, there is no “secs”, you use either “s” or “second”.
It is not clear how you go from 8 mini-platforms curves to the complete curve, as show in figure 3.
Please show a similar figure, but with the curves for each of the 8 platforms plus the sum of them.
Show them large enough for the ready to see the summing process.
L121: what is “a center curve can be calculated yielding the final GFz-curve”? Please show a plot of it and how you compute it. The right part of figure 1 is not very clear.
In figure 1 (or a new figure) please show a subject wearing the device.
Please check the company address and internet site, as the link shown appears as “for sale”.
Has is usual in Biomechanics, the gait cycle comprises the stance (support) and the swing (aerial) phases. Therefore, please update your manuscript to these common definitions. For instance, in figures 2 and 4 the horizontal axis is not labelled correctly.
Also usual is to abbreviate “Ground Reaction Force” by “GRF”, not “GF” as you do. Please change.
And it is common to normalize the GRF to the Body Weight (BW). In this case, the values do not depend on the weight of the subjects, as will be the case for values expressed in Newton: the 126kg subject will have values more than two times the values for the 48kg subject, just by the weight difference.
Please show the GRF values in BW.
The relation between the values of the peaks and valley to the gait speed will become apparent.
Although figure 4 is pertinent, it will be far more instructive to have a figure with the left and right feet simultaneously, i.e., put the curve of figure 3 in just one row. In this way the double support phases will be clearly seen, and their relation to the walking speed.
The results shown in figure 5 are not the same as some mentioned on the paragraph L175-182.
L267: “the gait was symmetric” how do you quantify this symmetry?
Somewhere in the text you must say why you do not use statistical parametric tests |
Indeed the word ”respond” was lost and caused this nonsense sentence.
We now use reviewer 2´s proposal!
This paragraph is now rewritten.
This is corrected.
This is corrected.
We prefer to use the term “club foot” only for foot deformities in patients with ICP. During rest feet were normal in our patients, only during walking or other muscular activities an abnormal position was observed. This is different to the “club foot” of ICP patients.
Instead of “laboratory finding” we use “analysis of blood and urine.
Reviewer 2 is absolutely right.
To reduce the number of tables the data of Table 1A and 1B are now included in the text. Since Reviewer 2 wanted to have more details on the “WD patients without two peaks” data of these patients are presented in the new Table 1. Table 1C is the new Table 2.
Each of the 2 tables is presented at its adequate place.
This is corrected.
This is corrected.
This is corrected.
Data of these 6 patients are presented
We now follow reviewer 2´s proposal.
This is corrected.
Reviewer 2 is absolutely right. It had to be 15.0 instead of 1.5.
This obvious mistake is corrected.
On the one hand we calculated the mean value for the parameter T1 and get a standard deviation across the entire cohort (interindividual variability of T1). On the other hand we calculated the intraindividual variability of T1 for each subject T1SD. Then we calculated the mean value of T1SD across the entire cohort and get another value (the mean intraindividual variability of T1). Standard deviation of T1 may be very different to mean T1SD.
We now follow reviewer 2´s proposal!
This is correct.
The complete curve is the sum of the 8 mini-platform curves.
The curves of the 8 sensors are not part of the output of the Ultraflex system. Therefore, it is impossible to produce such a figure.
Unfortunately, this is impossible.
The 8 mini-platforms are arranged in the x,y-plane and can be represented by 8 vectors. The vectors are weighted by their individual force value. The sum of these 8 vectors yields a vector representing the center of foot pressure to the ground (center curve). We do not use the information on the x,y-values of the center curve (called gait lines by the Infotronic® company) in the present paper and therefore have omitted this sentence in the revised manuscript.
An insert is presented with a subject wearing the device.
Indeed, the company who built out device seems to have difficulties, we do not have another address.
Reviewer 2 is absolutely right.
The labels of the horizontal axis in figures 2 and 4 are corrected.
We now use “GRF” as suggested by reviewer 2.
Normalizing of the data to BW implies a complete reanalysis of the data which is impossible in the moment.
This is impossible in the moment.
This may very well be. But in the moment the original data in Newton are not available to be rescaled by BW. We agree that GRF values in BW would probably improve the results.
We also agree that the simultaneous presentation of the curves of the left and right foot would have demonstrated the variability during a complete gait cycle more clearly. Unfortunately, the Ultraflex® device does not yield an output so that this analysis can be done easily. Furthermore, the temporal patterns are not the main topic of the present paper.
Reviewer 2 is absolutely right. The hint (Fig. 5) is given at the wrong place. That is corrected now.
We have compared the amplitude of peak 1 of the right and left leg. That is added now.
We have calculated correlations with the discrete MotS, N-MotS and TS. We therefore decided to use non-parametric tests throughout the manuscript. This is shortly mentioned in the “Statistics” section. |

Round 2
Reviewer 2 Report
Mildly impaired foot control in long term treated patients with Wilson’s disease
The manuscript improved from the previous version but still needs several improvements.
Please check carefully for occurrences of other mentions not “GRF” or “WD-patients”, as sometimes “GF” or “CD-patients” and other forms appear
Some capital “P” are still present, instead of lowercase “p”
L67: no paragraph here
Caption to figure 1: please put (left) description before (right) description
In P8-9 there still appears a Table 1A, 1B and 1C. it should be just Table 1 with the information on part 1A.
Part 1B should be table 2
Table 1 in P11 should be table 3
Table 2 in P13 should be table 4
L223, replace by: “number of steps (NUM) and step length needed”
L274, replace by: “earlier on average, and therefore”
L283-4, replace by: “Four of these 6 patients also were classified with abnormal gait during clinical examination.”
In the “Limitations of the study” please include a mention to the absence/impossibility to present GRF normalized to BW; include also a mention to the impossibility to have in the same plot the left and right “forces” that would show the double support phase.
Author Response
|
Reviewer 2 |
|
|
Mildly impaired foot control in long term treated patients with Wilson’s disease The manuscript improved from the previous version but still needs several improvements. Please check carefully for occurrences of other mentions not “GRF” or “WD-patients”, as sometimes “GF” or “CD-patients” and other forms appear Some capital “P” is still present, instead of lowercase “p” L67: no paragraph here Caption to figure 1: please put (left) description before (right) description In P8-9 there still appears a Table 1A, 1B and 1C.
it should be just Table 1 with the information on part 1A.
Part 1B should be table 2 Table 1 in P11 should be table 3
Table 2 in P13 should be table 4
L223, replace by: “number of steps (NUM) and step length needed” L274, replace by: “earlier on average, and therefore” L283-4, replace by: “Four of these 6 patients also were classified with abnormal gait during clinical examination.” In the “Limitations of the study” please include a mention to the absence/impossibility to present GRF normalized to BW; include also a mention to the impossibility to have in the same plot the left and right “forces” that would show the double support phase.
|
Thank you so much for great comments which helped us to improve the manuscript.
Unfortunately, the original table 1 remained in the revised version of the manuscript, although it should have been replaced by two other tables (see below). That caused a lot of problems. We apologize for that.
We have checked the manuscript for the presence of GF-curves or CD-patients.
We found several capital “P”. Sorry, that we did not eliminate them.
This is corrected.
This is improved.
Table 1 was totally modified. Following another reviewer who recommended to reduce the size of the paper we reduced the number of Tables to 2:
The content of Table 1A is now integrated into the text.
The content of Table 1B is now integrated into the text.
Table 1 now contains the data of the 6 patients without two peaks as recommended by reviewer 2.
Table 2 is the previous Table 1 C.
If reviewer 2 recommends to present 4 tables this can be modified within 1 day.
This is replaced.
This is replaced.
This is replaced.
Before we mention the advantage of the Ultraflex® system we now commend on some of the shortcomings of this system.
|
